# Predicting Regional Outbreaks of Hepatitis A Using 3D LSTM and Open Data in Korea

Kwangok Lee [1], Munkyu Lee [2] and Inseop Na [1,*]

1 National Program of Excellence in Software Centre, Chosun University, Gwangju 61452, Korea; csskwang@naver.com
2 Department of Electronic and Electrical Engineering, SungKyunKwan University, Suwon 16419, Korea; ansrb55@gmail.com
* Correspondence: ypencil@hanmail.net; Tel.: +82-62-230-6144

**Abstract:** In 2020 and 2021, humanity lived in fear due to the COVID-19 pandemic. However, with the development of artificial intelligence technology, mankind is attempting to tackle many challenges from currently unpredictable epidemics. Korean society has been exposed to various infectious diseases since the Korean War in 1950, and to overcome them, the six most serious cases in National Notifiable Infectious Diseases (NNIDs) category I were defined. Although most infectious diseases have been overcome, viral hepatitis A has been on the rise in Korean society since 2010. Therefore, in this paper, the prediction of viral hepatitis A, which is rapidly spreading in Korean society, was predicted by region using the deep learning technique and a publicly available dataset. For this study, we gathered information from five organizations based on the open data policy: Korea Centers for Disease Control and Prevention (KCDC), National Institute of Environmental Research (NIER), Korea Meteorological Agency (KMA), Public Open Data Portal, and Korea Environment Corporation (KECO). Patient information, water environment information, weather information, population information, and air pollution information were acquired and correlations were identified. Next, an epidemic outbreak prediction was performed using data preprocessing and 3D LSTM. The experimental results were compared with various machine learning methods through RMSE. In this paper, we attempted to predict regional epidemic outbreaks of hepatitis A by linking the open data environment with deep learning. It is expected that the experimental process and results will be used to present the importance and usefulness of establishing an open data environment.

**Keywords:** predicting; regional outbreaks; hepatitis A; deep learning; open data; big data; machine learning

## 1. Introduction

As we can see from the spread of COVID-19, SARS, and MERS, we can significantly reduce the number of victims if we can predict the epidemic. The reason why infectious diseases are considered "the existence of fear" in living things, including mankind, is because we do not know when, how and how they will occur [1–3].

Recently, many researchers have used the machine learning technique, a form of artificial intelligence, to obtain effective results in the prediction of changes in emotions or decision-making among people by data from social network systems, such as tweets on Twitter, posts on Facebook, and blogs [4,5]. Random Forest, Gradient Boost, Lasso, Ridge, Linear Regression, KNN, MLP, XG Boost, and Cat boost are commonly used for data prediction in machine learning techniques. Let us look at the pros and cons of some machine learning techniques. Linear regression offers advantages, such as simple implementation, easy understanding, quick training, and classification based on features. In the case of KNN, the advantages are ease of understanding and lower overheads in the adjustment of parameters. On the other hand, the disadvantages of linear regression include: its limitation to linear applications, its unsuitability to many real-life problems,

the default assumption of input error, and its assumption of independent features may not always be true. In the case of KNN, extra care required for the selection of K, and the cost of computation is high when working with large datasets.

Recently, various disease prediction studies have been published. Santos, Carlos, and Matos studied influenza in 2014, but they only considered Portugal in their proposed work [6]. In 2015, Grover, Sangeeta, and Aujla processed data using tweets for swine flu [7]. In 2017, McGough and Sarah F studied zika virus, and they only predicted one parameter for forecasting [8]. In 2018, Nair, Lekha R., Sujala D. Shetty, and Siddhanth D. Shetty studied heart disease; however, they did not do so under the category of epidemics, so their study needed to be linked with a health care service provider in order to work in real time [9]. In 2019, Maurice and Nduwayezu studied malaria; their study was limited to Nigeria only [10]. In 2020, Petropoulos, Fotios, and Makridakis worked on COVID-19, but they did not use machine learning [11].

In Korea, there are six cases of National Notifiable Infectious Diseases (NNIDs) at category I infection according to the definition established in 1954, as shown in Table 1. Recently, rates of cholera, typhoid fever, paratyphoid fever, shigellosis, and enterohemorrhagic Escherichia coli have been low in Korea. Typhoid fever, cholera, and shigellosis in particular were highly prevalent in the 1960s. According to the analysis of the nation's hepatitis A antibody retention rate for the 10 years between 2005 and 2014, 7 out of 10 infected people are in their 30s and 40s, and hepatitis A prevention measures for this age group are necessary. In the past 10 years, Korea has taken the openness of public data as a national indicator and has been opening up various daily data, such as population data, meteorological observation data, water quality data, and air quality data. For this reason, using stable and high-accuracy deep learning technology, we have been able to verify the relationship between diseases and the public data on daily life collected over many years.

**Table 1.** Prevalence of National Notifiable Infectious Category I Diseases in Korea (restructured based on [12,13]).

| Year / Disease | 1954–1959 | 1960–1969 | 1970–1979 | 1980–1989 | 1990–1999 | 2000–2009 | 2010–2013 | 2014 | 2015 | 2016 | 2017 |
|---|---|---|---|---|---|---|---|---|---|---|---|
| *Cholera* | 0 | 1972 | 206 | 145 | 196 | 210 | 14 | 0 | 0 | 4 | 5 |
| *Typhoid fever* | 5398 | 40,790 | 13,018 | 2481 | 3012 | 2198 | 566 | 251 | 121 | 121 | 128 |
| *Paratyphoid fever* | 193 | 440 | 64 | 172 | 164 | 795 | 223 | 37 | 44 | 56 | 73 |
| *Shigellosis* | 1004 | 2705 | 1703 | 534 | 3368 | 6986 | 783 | 110 | 88 | 113 | 111 |
| *Enterohemorrhagic Escherichia coli* | - | - | - | - | - | 431 | 246 | 111 | 71 | 104 | 139 |
| *Viral hepatitis A* | - | - | - | - | - | - | 7585 | 1307 | 1804 | 4679 | 4429 |

Hence, in this paper, we aim to minimize the costs and damages involved in the prevention of epidemic outbreaks by predicting regional outbreaks of hepatitis A by using publicly available data in Korea and recent machine learning algorithms.

## 2. Prediction System of Hepatitis A

To predict hepatitis A, we conducted a two-phase approach, as shown in Figure 1.

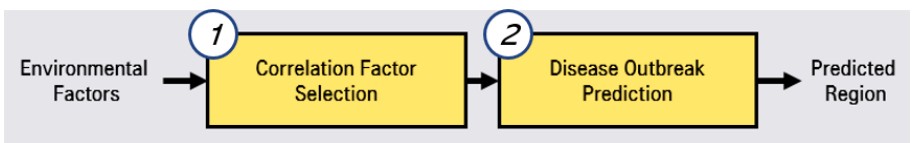

**Figure 1.** Two-phase approach of prediction system for hepatitis A.

The first step is correlated factor selection for learning for the prediction model. In this correlated factor selection step, we separate irrelevant factors from environmental factors

through statistical analysis. The second step is disease outbreak prediction through LSTMs (long short-term memory networks) [14,15]. In this phase of the prediction, we preprocess the selected correlated factors and predictions by using LSTMs.

### 2.1. Correlated Factor Selection

In this correlated factor selection, we conduct data gathering, data preprocessing, and statistical analysis, as shown in Figure 2. First, we perform web crawling to gather the open data for each region in Korea by studying open data sites in Korea.

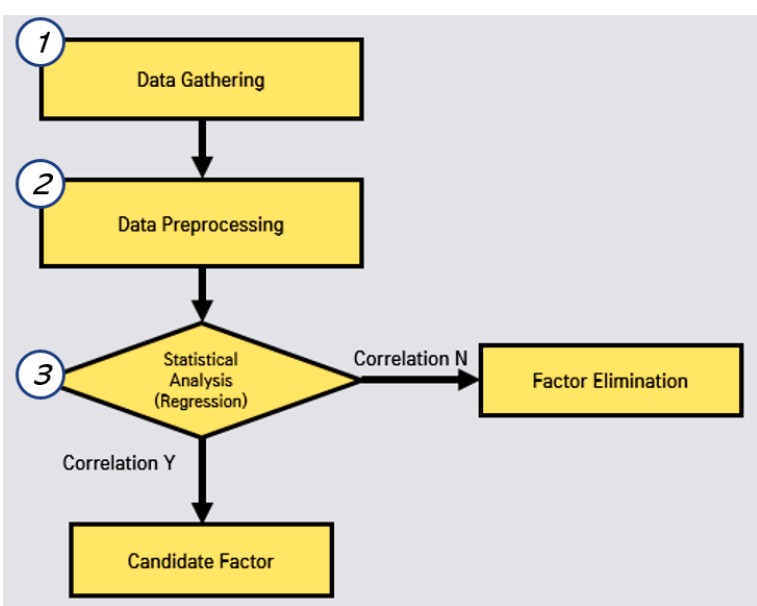

**Figure 2.** Process of correlated factor selection.

1. Patient information: KCDC (Korea Centers for Disease Control & Prevention), http://www.cdc.go.kr;
2. Water Environment Information: NIER (National Institute of Environmental Research), http://water.nier.go.kr/publicMain/mainContent.do;
3. Weather Information: KMA (Korea Meteorological Agency), https://data.kma.go.kr;
4. Population Information: Public Open Data Portal, https://www.data.go.kr/;
5. Air Pollution Information: KECO (Korea Environment Corporation) AirKorea, https://www.airkorea.or.kr.

Second, we perform data preprocessing for the missing values, the regulations of individual regions in Korea. Third, we perform the evaluation of the correlation between the disease (hepatitis A) and each environmental factor. In this evaluation, we eliminate the non-related factors. Subsequently, we can obtain the candidate factors to predict the outbreak.

### 2.2. Disease Outbreak Prediction with Hepatitis A by Regression Analysis

In this disease outbreak prediction, we conduct the two steps, data preprocessing and LSTMs by using selected correlated factor (candidate factor), as shown in Figure 3. In the preprocessing step, we reorganize the data by living area, feature scaling from 0 to 1. In the prediction by LSTMs step, we calculate that RMSE (Root Mean Square Error) [16] for Random Forest [17], Gradient Boosting Regression [18], Lasso [19], Ridge [20], Linear Regression [21], K-Neighbors Regression, MLP (Multi-Layer Perceptron) Regression [22], XGB Regression, and Cat Boost Regression. These RMSE evaluation results are used for the determination of hyper-parameter adjustment and optimal algorithm selection.

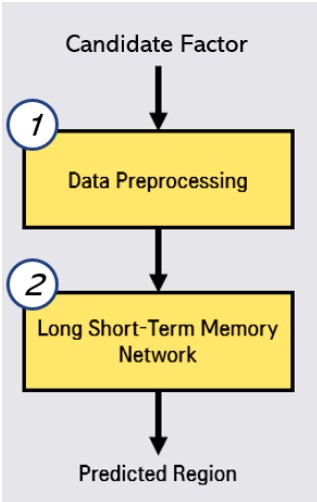

**Figure 3.** Process of outbreak region prediction for hepatitis A.

## 3. Experimental Results

### 3.1. Correlated Factor Selection

We gather the data from the websites that mention 'A. Correlated Factor Selection' through web crawling, as shown in Table 2.

**Table 2.** Examples of web crawling data.

| | Date | Province | County | Cholera | Typhoid | Paratyphoid | Shigellosis | Enterohemorrhagic Escherichia Coli | Hepatitis A |
|---|---|---|---|---|---|---|---|---|---|
| 22972 | 2016-09 | Gangwon | Yangyang | 0.000000000 | 0.000000000 | 0.000000000 | 0.000000000 | 0.000000000 | 0.000000000 |
| 22973 | 2016-09 | Gangwon | Yeongwol | 0.000000000 | 0.000000000 | 0.000000000 | 0.000000000 | 0.000000000 | 0.000000000 |
| 22974 | 2016-09 | Gangwon | Wonju-si | 0.000000000 | 0.000000000 | 0.000000000 | 0.000000000 | 0.000000000 | 1.790000000 |
| 22975 | 2016-09 | Gangwon | Inje | 0.000000000 | 0.000000000 | 0.000000000 | 0.000000000 | 0.000000000 | 0.000000000 |
| 22976 | 2016-09 | Gangwon | Jeongseon | 0.000000000 | 0.000000000 | 0.000000000 | 0.000000000 | 0.000000000 | 0.000000000 |
| 22977 | 2016-09 | Gangwon | Cheorwon | 0.000000000 | 0.000000000 | 0.000000000 | 0.000000000 | 0.000000000 | 0.000000000 |
| 22978 | 2016-09 | Gangwon | Chuncheon-si | 0.000000000 | 0.000000000 | 0.000000000 | 0.000000000 | 0.000000000 | 1.430000000 |

Measurement data that are missing for various reasons are called missing values. Missing values are displayed as None, NaN, or blank in the program, and a dataset with many such missing values greatly affects the quality of the statistical prediction in the model. In particular, in machine learning models, all input values are assumed to be meaningful values, so missing values further affect the quality of the model. Rubin [23] classified missing data problems into three categories, which are missing completely at random (MCAR), missing at random (MAR), and not missing at random (NMAR). If the probability of being missing is the same for all cases, then the data are said to be MCAR. If the probability of being missing is the same only within groups defined by the observed data, then the data are MAR. If neither MCAR nor MAR holds, then the probability is MNAR. The methods of dealing with missing values are cross-sectional data, consisting of observation values viewed at one point in time for each item, and panel data (longitudinal data), consisting of observation values of multiple objects from multiple viewpoints using time series data. Methods commonly used for cross-sectional data include removing missing values, the imputation of mean or median values, the imputation of the most frequent values or 0 or specific constants, the imputation of K-NN, the MICE (Multivariate Imputation by Chained Equation) imputation method, and imputation using deep learning.

In this study, we used deep learning-based imputation, which is currently widely used; it is more accurate than other methods and has the ability to process a feature encoder.

When there are too many missing value items, corresponding items are removed. We measured the missing values using a Random Forest regressor, and both the previous subsequent five the missing values were used as training data. We set the estimator to 50 and the max depth to 4 to prevent overfitting because there was little training data.

We removed the missing value, as shown in Table 3. We marked the missing value as '*' to represent the blank information, as shown in Table 3 (upper). We replaced the missing values with new values according to the missing values policy, as shown in Table 3 (lower).

**Table 3.** Replace processing for missing value (upper: original data, lower: replaced missing value).

| Total Nitrogen | Total Phosphorus | TOC | Mercury |
|---|---|---|---|
| 7.346666667 | 0.094 | 0.366666667 | 16.16666667 |
| 7.615333333 | 0.106333333 | 0.4 | 14.13333333 |
| 9.486333333 | 0.090333333 | 0.4 | 16.6 |
| 9.226 | 0.091666667 | 0.4 | 15.96666667 |
| 9.023666667 | 0.112 | 0.433333333 | 15.86666667 |
| 6.408 | 0.087333333 | 0.466666667 | 14.7 |
| 7.346 | 1.114666667 | * | 14.4 |
| 7.738333333 | 0.127 | 0.466666667 | 14.16666667 |
| 4.906666667 | * | 0.5 | * |
| * | 0.085333333 | * | 17.2 |
| 8.348333333 | 0.088333333 | * | 17.66666667 |
| * | 0.00333333 | * | 15.9 |
| **Total Nitrogen** | **Total Phosphorus** | **TOC** | **Mercury** |
| 7.346666667 | 0.094 | 0.366666667 | 16.16666667 |
| 7.615333333 | 0.106333333 | 0.4 | 14.13333333 |
| 9.486333333 | 0.090333333 | 0.4 | 16.6 |
| 9.226 | 0.091666667 | 0.4 | 15.96666667 |
| 9.023666667 | 0.112 | 0.433333333 | 15.86666667 |
| 6.408 | 0.087333333 | 0.466666667 | 14.7 |
| 7.346 | 0.114666667 | *0.466666667* | 14.4 |
| 7.738333333 | 0.127 | 0.466666667 | 14.16666667 |
| 4.906666667 | *0.081333333* | 0.5 | *13.66666667* |
| *8.156666667* | 0.085333333 | *0.5* | 17.2 |
| 8.348333333 | 0.088333333 | *0.5* | 17.66666667 |
| *7.079333333* | 0.100333333 | *0.533333333* | 15.9 |

Italic, underline and bold number: new values according to the missing values policy; *: missing value.

We performed the data regulation and region regulation for the monthly data as the mean of the monthly measured data, the integration for the region as living area and the correlated environment data, as shown in Figure 4. Figure 4a (left) shows the original data and Figure 4a (right) shows the mean of the monthly data. The publicly available data includes water quality measurement data that does not exist at a specific time due to problems such as the installation of measurement sensors. To solve this problem, we recombined the regions based on the living area, as shown in Figure 4b, and divided them into eight areas. Each color was arbitrarily selected as a color that could clearly distinguish the region.

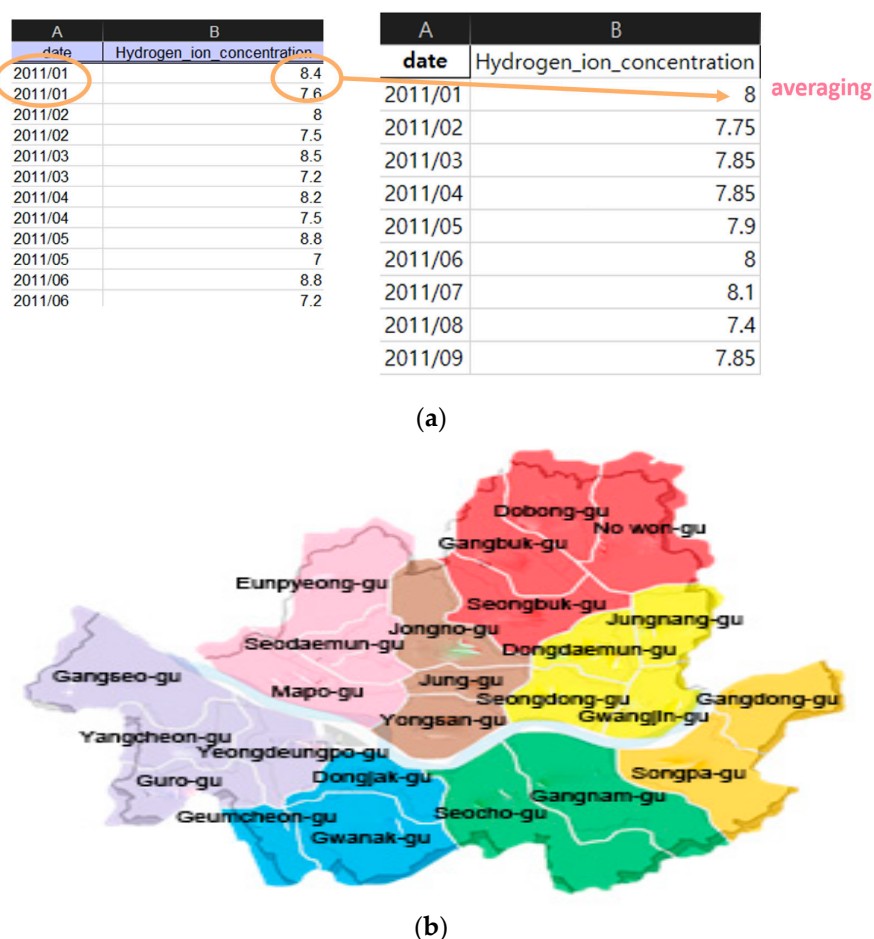

(a)

(b)

**Figure 4.** Regulation of data (**a**) left: original data, upper right: mean of monthly data, (**b**) integrated living area.

We then adjusted the number of epidemic outbreaks to the number of outbreaks per 100,000 population in order to measure the same conditions across different regions, as shown in Table 4.

**Table 4.** Epidemic outbreaks to the number of outbreaks per 100,000 population.

| Date | Area | No. of Outbreaks | Population | No. of Outbreaks per 100,000 Population |
|---|---|---|---|---|
| 2016-01 | Kangwon | 1 | 1,549,193 | 0.059888984 |
| 2016-01 | Gyeonggi | 200 | 12,536,474 | 1.593915342 |
| 2016-01 | Gyeongnam | 81 | 3,364,764 | 2.398269409 |
| 2016-01 | Kyongbuk | 10 | 2,701,160 | 0.374661333 |
| 2016-01 | Gwangju | 2 | 1,472,802 | 0.135795579 |
| 2016-01 | Daegu | 4 | 2,487,823 | 0.148598832 |
| 2016-01 | Daejeon | 2 | 1,518,024 | 0.160783143 |

We used multiple regression analysis to verify reliable factors in the relationship between hepatitis A and environmental factors. We validated the goodness of fit of the model by using the R-squared value, as shown in Table 5. We obtained an R-squared value of 0.7054. We present the positive correlations for the COD (Chemical Oxygen Demand) values, total coliform count, total dissolved nitrogen, daily precipitation, and PM10 (particulate matter) in italic bold blue characters, and an $ indicator after the item

name, in Table 5. The negative correlations for the TOC (Total Organic Carbon) values, number of Fecal E. coliform, monthly precipitation, and so2 are presented in italic, bold, underlined red characters, with a % indicator after the item name, in Table 5. Figure 5 shows the statistical results of the linear hypothesis between hepatitis A and the environmental factors. As a result of the test, the differences between the two groups were interpreted as statistically significant.

**Table 5.** Multiple regression analysis results of each environmental factor and hepatitis A.

| Items | Estimate Std. | Error | t Value | Pr(>|t|) |
|---|---|---|---|---|
| (Intercept) | 1.084 | $4.783 \times 10^{-1}$ | 2.267 | 0.02394 * |
| pH | $-2.461 \times 10^{-2}$ | $2.861 \times 10^{-2}$ | −0.860 | 0.39017 |
| Dissolved Oxygen | $-2.812 \times 10^{-3}$ | $1.007 \times 10^{-2}$ | −0.279 | 0.78009 |
| BOD | $-1.008 \times 10^{-2}$ | $1.414 \times 10^{-2}$ | −0.713 | 0.47645 |
| COD $ | $4.032 \times 10^{-2}$ | $1.664 \times 10^{-2}$ | 2.423 | 0.01587 * |
| Suspended Solid | $1.642 \times 10^{-3}$ | $2.860 \times 10^{-3}$ | 0.574 | 0.56611 |
| Total Nitrogen | $-1.448 \times 10^{-1}$ | $4.887 \times 10^{-2}$ | −2.963 | 0.00324 ** |
| Total Phosphorus | $6.142 \times 10^{-1}$ | $2.755 \times 10^{-1}$ | 2.229 | 0.02639 * |
| TOC % | $-5.964 \times 10^{-2}$ | $1.233 \times 10^{-2}$ | −4.835 | $1.94 \times 10^{-6}$ *** |
| Water Temperature | $1.208 \times 10^{-3}$ | $8.672 \times 10^{-3}$ | 0.139 | 0.88926 |
| Conductivity | $-3.274 \times 10^{-4}$ | $1.678 \times 10^{-4}$ | −1.951 | 0.05183. |
| Total Coliforms $ | $3.369 \times 10^{-7}$ | $1.348 \times 10^{-7}$ | 2.499 | 0.01287 * |
| Dissolved total Nitrogen $ | $1.454 \times 10^{-1}$ | $5.331 \times 10^{-2}$ | 2.727 | 0.00670 ** |
| 'Ammonia Nitrogen' | $2.935 \times 10^{-2}$ | $2.693 \times 10^{-2}$ | 1.090 | 0.27636 |
| 'Nitrate Nitrogen' | $1.170 \times 10^{-2}$ | $2.392 \times 10^{-2}$ | 0.489 | 0.62494 |
| Dissolved total Phosphorus | $-8.784 \times 10^{-3}$ | $2.565 \times 10^{-2}$ | −0.342 | 0.73224 |
| Phosphate Phosphorus | $-5.387 \times 10^{-1}$ | $3.614 \times 10^{-1}$ | −1.490 | 0.13695 |
| Chlorophyll | $3.387 \times 10^{-3}$ | $2.001 \times 10^{-3}$ | 1.693 | 0.09131. |
| Fecal E. coliform count % | $-5.887 \times 10^{-6}$ | $1.922 \times 10^{-6}$ | −3.063 | 0.00235 ** |
| Average temperature | $-1.371 \times 10^{-1}$ | $2.557 \times 10^{-2}$ | −5.361 | $1.44 \times 10^{-7}$ *** |
| Highest temperature | $6.476 \times 10^{-3}$ | $8.274 \times 10^{-3}$ | 0.783 | 0.43432 |
| Lowest temperature | $2.760 \times 10^{-2}$ | $8.681 \times 10^{-3}$ | 3.179 | 0.00160 ** |
| Average relative humidity | $-5.186 \times 10^{-3}$ | $5.537 \times 10^{-3}$ | −0.937 | 0.34957 |
| Monthly precipitation % | $-1.636 \times 10^{-3}$ | $3.560 \times 10^{-4}$ | −4.596 | $5.86 \times 10^{-6}$ *** |
| Daily maximum precipitation | $6.989 \times 10^{-3}$ | $1.218 \times 10^{-3}$ | 5.737 | $1.97 \times 10^{-8}$ *** |
| Small total evaporation | $-5.467 \times 10^{-3}$ | $1.210 \times 10^{-3}$ | −4.518 | $8.33 \times 10^{-6}$ *** |
| Average wind speed | $8.834 \times 10^{-2}$ | $7.598 \times 10^{-3}$ | 1.163 | 0.24572 |
| Average amount of cloud | $2.856 \times 10^{-2}$ | $2.317 \times 10^{-2}$ | 1.232 | 0.21856 |
| The most serious theory | $-1.148 \times 10^{-2}$ | $6.809 \times 10^{-3}$ | −1.685 | 0.09273. |
| Average ground temperature | $1.115 \times 10^{-1}$ | $2.041 \times 10^{-2}$ | 5.463 | $8.48 \times 10^{-8}$ *** |
| so2 % | $-1.434e \times 10^{+2}$ | $2.694e \times 10^{+1}$ | −5.324 | $1.74 \times 10^{-7}$ *** |
| no2 | $-5.512 \times 10^{-3}$ | 5.934 | −0.093 | 0.92605 |
| o3 | −9.996 | 5.941e | −1.683 | 0.09326. |
| co | $-3.551 \times 10^{-1}$ | $3.715 \times 10^{-1}$ | −0.956 | 0.33978 |
| pm10 $ | $1.959 \times 10^{-2}$ | $2.547 \times 10^{-3}$ | 7.690 | $1.27 \times 10^{-13}$ *** |

Significance code and p-value: ***: [0, 001] ; **: (0.001, 0.01] ; *: (0.01, 0.05] ; .: (0.05, 0.1]; : (0.1, 1]. Residual standard error: 0.866 on 548 degrees of freedom; Multiple R-squared: 0.7357, adjusted R-squared: 0.7054; F-statistic: 24.22 on 63 and 548 DF, *p*-value: < $2.2 \times 10^{-16}$. Red marked text and underline text: negative correlations; Blue marked text: positive correlations.

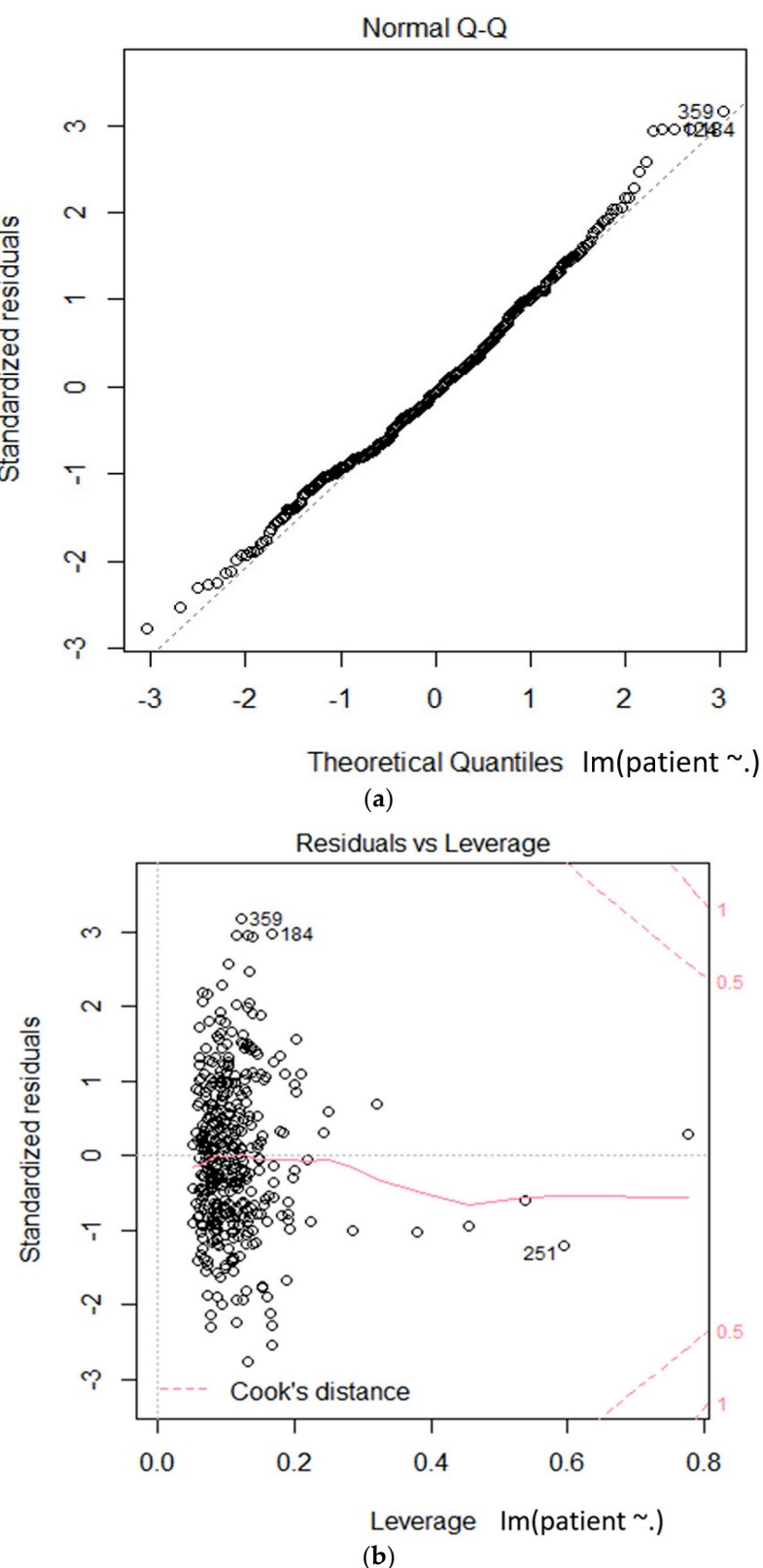

**Figure 5.** Linear model based statistical test on the relationships between hepatitis A and environmental factors (**a**): theoretical quantiles vs standardized residuals; (**b**): leverage vs standardized residuals.

Figure 6 shows that the results of validation of correlation coefficient between environmental factors using heatmap Figure 6 shows that some environmental predictors of the model used in the regression analysis we used have low correlations with other envi-

ronmental predictors in the correlation coefficient between hepatitis A and environmental factors. Therefore, it was verified that the data analysis did not show a negative effect. The 29 environmental factors used to correlate with hepatitis A patients information are hydrogen ion concentration, dissolved oxygen, BOD, COD, suspended solids, total nitrogen, total phosphorus, TOC, mercury, electrical conductivity, total coliform bacteria, dissolved total nitrogen, ammonia nitrogen, Acid nitrogen, dissolved total phosphorus, phosphate phosphorus, chlorophyll, E. coli bacteria, average temperature, maximum temperature, minimum temperature, average relative humidity, monthly precipitation, highest daily precipitation, small total evaporation, average wind speed, average cloud quantity, deep snow, average ground Temperature.

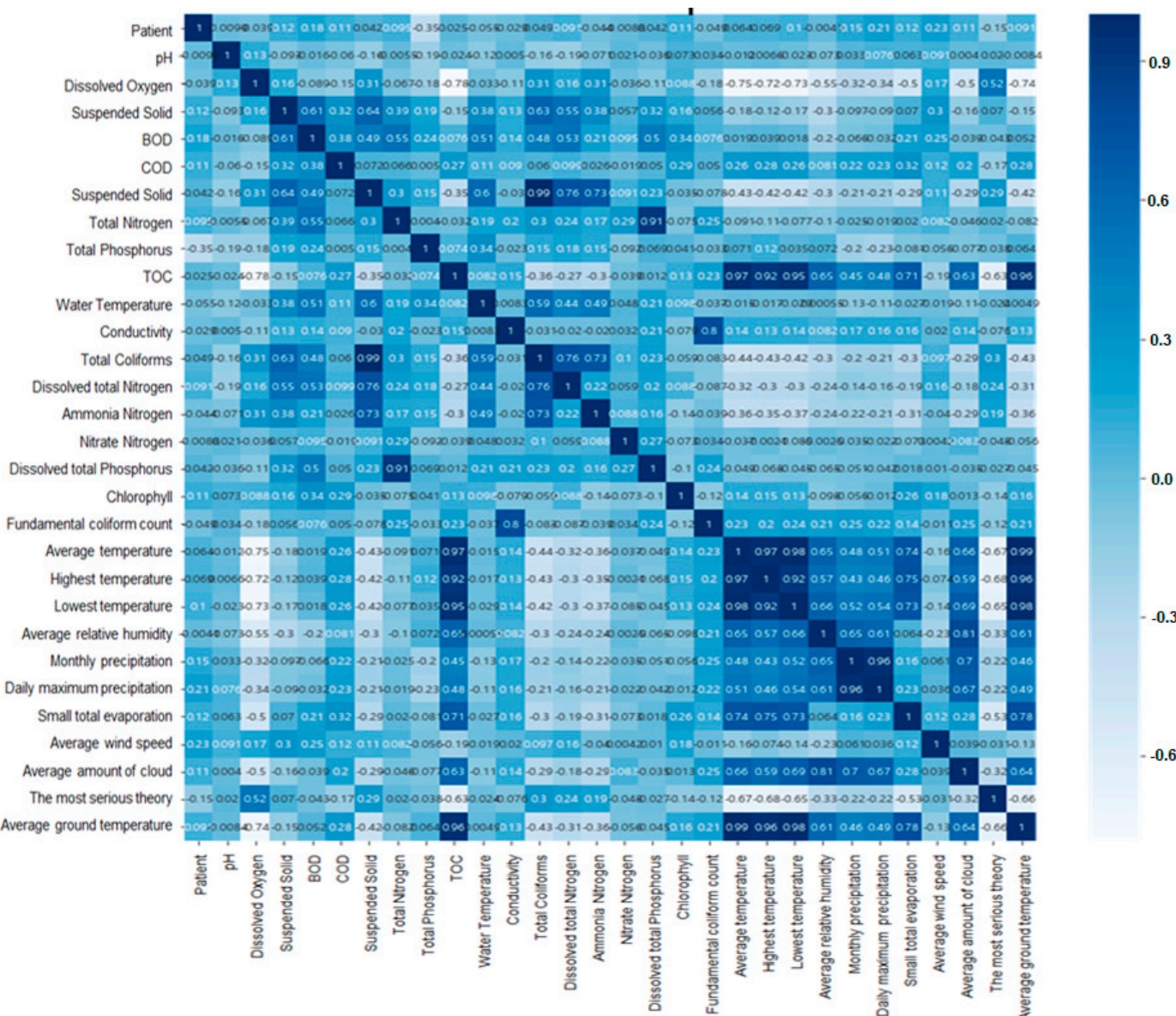

**Figure 6.** Validation of correlation coefficient between environmental factors using heatmap.

### 3.2. Outbreak Region Prediction of Hapatitis A

Through the correlated factor selection process, we integrated patient information, water environment information, weather information, population information, and air pollution information, and refined the data per 100,000 population to obtain the results shown in Table 6. We removed data without patient information or relevant local informa-

tion during this process. The data obtained were divided into 17 areas across the country, with relevance for 50 items, and Seoul was recombined into eight areas based on living standards. The data obtained are 613 national data from 2016 to 2018 and 769 Seoul data from 2011 to 2018.

**Table 6.** Fifty integrated factors (patient information, region information, environmental factors).

| Items | Value 1 | Value 2 | Value 3 | Value 4 | Value 5 | Value 6 |
|---|---|---|---|---|---|---|
| Patient | 0.059889 | 1.5939153 | 2.3982694 | 0.3746613 | 0.1357956 | 0.1485988 |
| Area | Kangwon | Gyeonggi | Gyeongnam | Kyongbuk | Gwangju | Daegu |
| Population | 1549193 | 12536474 | 3364764 | 2701160 | 1472802 | 2487823 |
| pH | 7.85 | 8.03 | 7.7 | 7.8588235 | 7.3583333 | 7.805814 |
| Dissolved Oxygen | 12.492308 | 12.91 | 10.766667 | 13.772059 | 13.925 | 14.05 |
| BOD | 3.7923077 | 3.04 | 0.5 | 1.6132353 | 2.1083333 | 1.2976744 |
| COD | 6.5038462 | 6.02 | 0.4666667 | 4.4455882 | 4.925 | 3.8709302 |
| Suspended Solid | 5.4961538 | 6.37 | 0.5333333 | 6.1 | 7.1 | 4.4209302 |
| Total Nitrogen | 8.9094231 | 3.6515 | 4.9066667 | 4.0757647 | 4.3966667 | 2.6079186 |
| Total Phosphorus | 0.1218077 | 0.059 | 0.0813333 | 0.0725735 | 0.09 | 0.0396628 |
| TOC | 3.5 | 3.12 | 0.5 | 2.5147059 | 3.3083333 | 2.2813953 |
| Water Temperature | 3.9423077 | 4.07 | 13.666667 | 3.3529412 | 4.4666667 | 4.0860465 |
| Conductivity | 614.61538 | 4778.8 | 157.33333 | 310.82353 | 375.83333 | 339.83721 |

Table 7 shows the normalized data by data scaling. We use the min–max normalization for rescaling the features. Min–max normalization consists in rescaling the range of features to scale the range in [0,1]. The Equation (1) for a min-max of [0,1] is given as:

$$x' = \frac{x - \min(x)}{\max(x) - \min(x)} \tag{1}$$

**Table 7.** Example of data scaling (upper: original data, lower: normalized data).

| Population | pH | Dissolved Oxygen | BOD | COD | Suspended Solid | Total Nitrogen |
|---|---|---|---|---|---|---|
| 3511974 | 7.867978 | 10.591573 | 3.964045 | 6.862360 | 12.588202 | 4.982775 |
| 2701238 | 7.822059 | 11.044118 | 2.661765 | 6.372059 | 12.775000 | 3.625824 |
| 13090648 | 7.756471 | 13.115882 | 2.160000 | 4.574706 | 5.106471 | 6.055618 |
| Population | pH | Dissolved Oxygen | BOD | COD | Suspended Solid | Total Nitrogen |
| 3511974 | 0.638519 | 0.446706 | 0.410341 | 0.529616 | 0.207408 | 0.400355 |
| 2701238 | 0.621617 | 0.491382 | 0.272046 | 0.490444 | 0.210587 | 0.248327 |
| 13090648 | 0.597474 | 0.695910 | 0.218761 | 0.346847 | 0.080091 | 0.520553 |

Table 7 (upper) represents the original data before scaling and Table 7 (lower) represents the data normalized by data scaling. In this process, we produce the same scale data for training and testing.

We chose the optimal model to be used for the LSTM network. Nine algorithms were tested, including Random Forest, Gradient Boost, Lasso, Ridge, Linear Regression, KNN, MLP, XG Boost, and Cat boost. Table 8 shows the comparison results for the nine algorithms to choose the candidate for tuning the hyper-parameters. We used the RMSE (Root Mean Square Errors) to compare the algorithm. According to the experimental

results, Gradient Boost, Cat Boost, and Random Forest were selected for tuning the hyper-parameters. After tuning the parameters, the best optimal algorithm was Gradient Boost, whose value changed from 0.077935 to 0.0759682. We estimated the optimal parameter using Grid Search CV [24] for Gradient Boost, and modified the learning rate from the default value of 0.1 to 0.075, the N_estimators from the default value of 100 to 200, and the max_depth from the default value of 3 to 4. Grid search CV is a function provided by sklearn that automatically learns the number of cases that can be made with the values by entering the desired hyper-parameter and numerical range. Furthermore, it calculates the best-performing parameter as the final output based on the evaluation index (in this paper we used MSE) set by the user, based on the learned data [24].

**Table 8.** RMSE Comparison results for the nine algorithms.

|  | Algorithm | RMSE (Original) | RMSE (Tuning) |
|---|---|---|---|
| 0 | RandomForestRegressor | 0.081008 | - |
| 1 | *GradientBoostingRegressor* | 0.077935 | 0.075904 |
| 2 | Lasso | 0.147475 | - |
| 3 | Ridge | 0.085332 | - |
| 4 | LinearRegression | 0.086475 | - |
| 5 | KNeighborsRegressor | 0.110483 | - |
| 6 | *MLPRegressor* | 0.096595 | - |
| 7 | XGBRegressor | 0.078657 | - |
| 8 | CatBoostRegressor | 0.081142 | - |

The tests were conducted in one area of Seoul, the training data used were from 2016 to March 2018, and the validation data used were from April to October 2018. To perform the predictions, the tests were conducted using data from November and December 2018. The epidemic of hepatitis A is shown in Figure 7. The blue line is the training data, the orange line is the validation data, the green line is the test data. Figure 7 visually presents the selection of training data, validation data, and test data within the time series data, including the change in the number of hepatitis A patients.

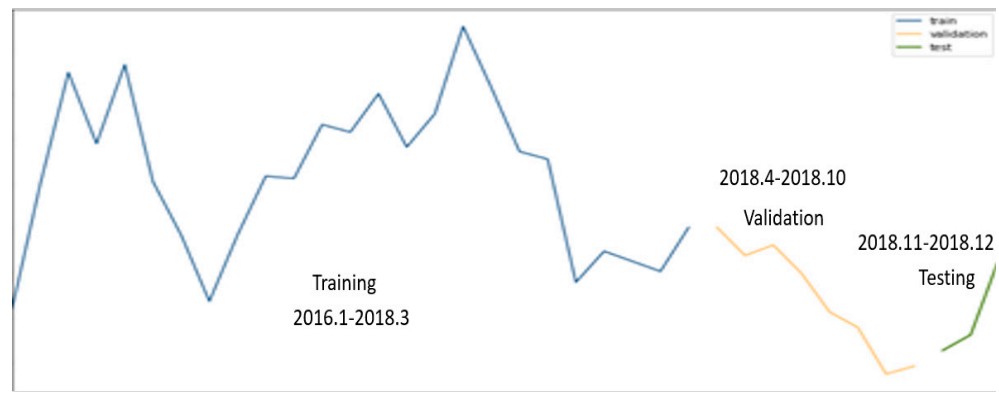

**Figure 7.** Visualization of epidemic of hepatitis A dataset (training, validation, test data).

We transformed the 2D data into 3D data, as shown in Figure 8. The 2D data comprised a number of features and samples. The 3D data comprised a number of features, samples, and time steps. In order to predict the $y\_t + 1$ time point using the LSTM, a total of six time steps was used from the $y\_t$ time point to the past $y\_t-5$, as shown in Table 9. In our model, we use the sequential model, the LSTM layer, and the Dense layer. The optimizer is RMSprop (Root Mean Square propagation) and the loss function is MSE (Mean Square

Error). RMSProp prevents the learning rate from dropping too close to zero by reflecting only the information of the new slope, rather than adding all the previous slopes uniformly. MSE is the most commonly used regression loss function. MSE is the sum of squared distances between the target variable and the predicted values. In order to process small data, the batch size was set to 2.

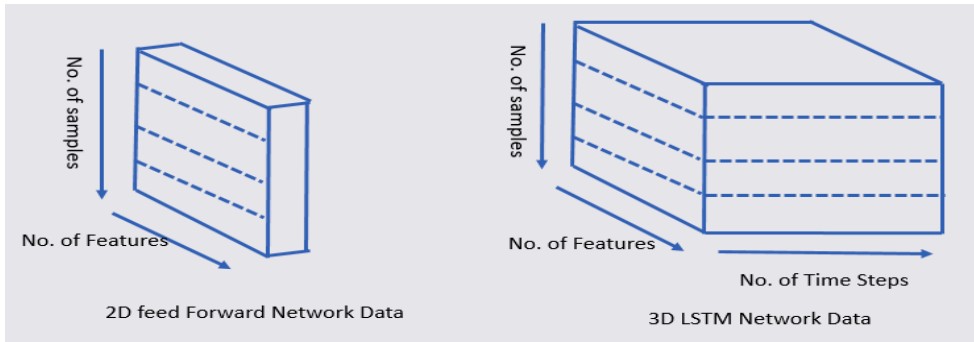

**Figure 8.** Transformation of 2D feed-forward data into 3D LSTM data.

**Table 9.** Serial data for LSTM.

| | y | X | | | | | |
|---|---|---|---|---|---|---|---|
| **load** | **y_t + 1** | **y_t − 5** | **y_t − 4** | **y_t − 3** | **y_t − 2** | **y_t − 1** | **y_t** |
| 0.00 | 0.44 | nan | nan | nan | nan | nan | 0.00 |
| 0.44 | 0.84 | nan | nan | nan | nan | 0.00 | 0.44 |
| 0.84 | 0.59 | nan | nan | nan | 0.00 | 0.44 | 0.84 |
| 0.59 | 0.86 | nan | nan | 0.00 | 0.44 | 0.84 | 0.59 |
| 0.86 | 0.45 | nan | 0.00 | 0.44 | 0.84 | 0.59 | 0.86 |
| 0.45 | 0.26 | 0.00 | 0.44 | 0.84 | 0.59 | 0.86 | 0.45 |
| 0.26 | 0.03 | 0.44 | 0.84 | 0.59 | 0.86 | 0.45 | 0.26 |
| 0.03 | 0.26 | 0.84 | 0.59 | 0.86 | 0.45 | 0.26 | 0.03 |
| 0.26 | 0.47 | 0.59 | 0.86 | 0.45 | 0.26 | 0.03 | 0.26 |
| 0.47 | 0.46 | 0.86 | 0.45 | 0.26 | 0.03 | 0.26 | 0.47 |
| 0.46 | 0.65 | 0.45 | 0.26 | 0.03 | 0.26 | 0.47 | 0.46 |

Background: y:y_t + 1 time point for prediction, X:from y_t time point to the past y_t − 5.

We conduct 15 epochs for learning. Early stopper is used to halt the training of the LSTMs at the right time to avoid overfitting and underfitting the model.

For this paper, because of the amount of data used was not large, we applied the early stopping algorithm to prevent overfitting. Figure 9 shows the comparison results of the predicted and actual values for one area of Seoul.

Figure 10 shows the prediction results for the epidemic of hepatitis A in Seoul. We used the training data (from January 2016 to July 2018) and the test data (August 2018) on the eight recombined areas of Seoul. The circle symbol is the actual data and the start mark is the predicted data for each area.

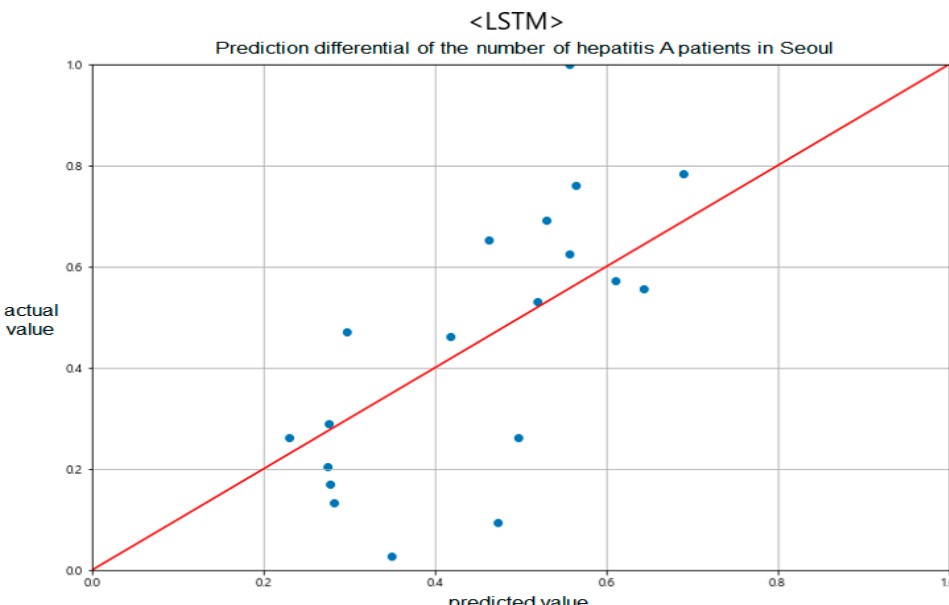

**Figure 9.** Prediction differential of the number of hepatitis A patients in Seoul between predicted and actual values.

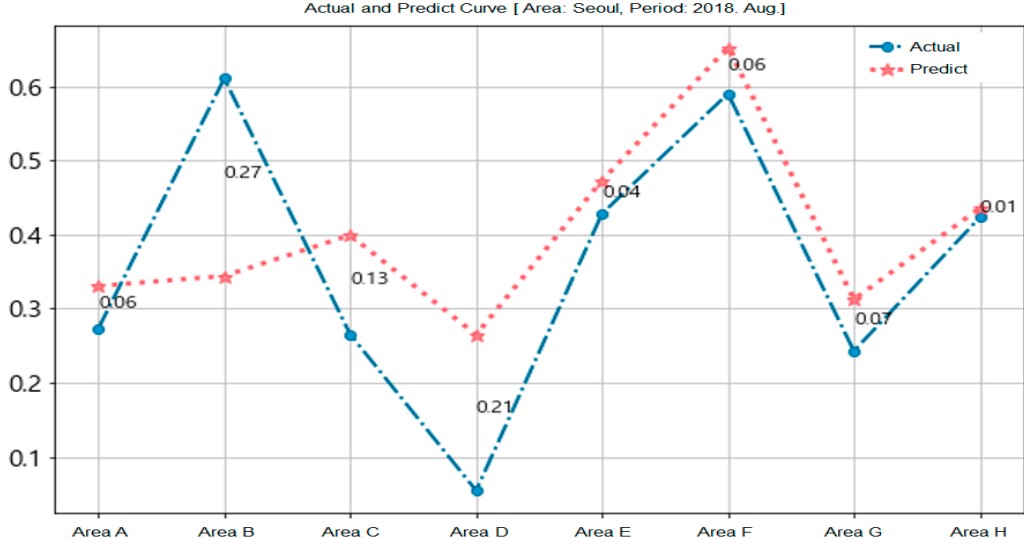

**Figure 10.** Prediction of epidemic of hepatitis A in Seoul.

Areas B and D demonstrate many differences between forecasts and measurements because the weather and air pollution information used in the forecasts were not measured in a specific area, but rather across Seoul. This is another potential reason for the error that occurred when forcibly setting the eight recombined areas as the district area of Seoul.

Figures 11 and 12 show the national 17-area prediction of the epidemic of hepatitis A in Korea for each local government unit. We used the training data (from January 2016 to November 2018) and the test data (December 2018). The blue circle symbol is the actual data and the red star symbol is the predicted data for each area in Figure 11.

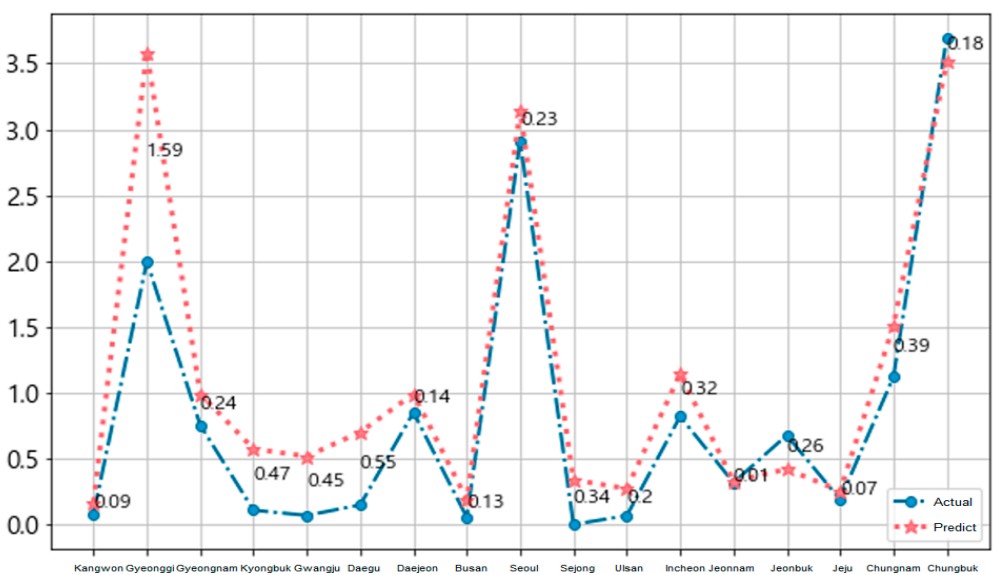

**Figure 11.** National prediction of epidemic of hepatitis A per local government unit.

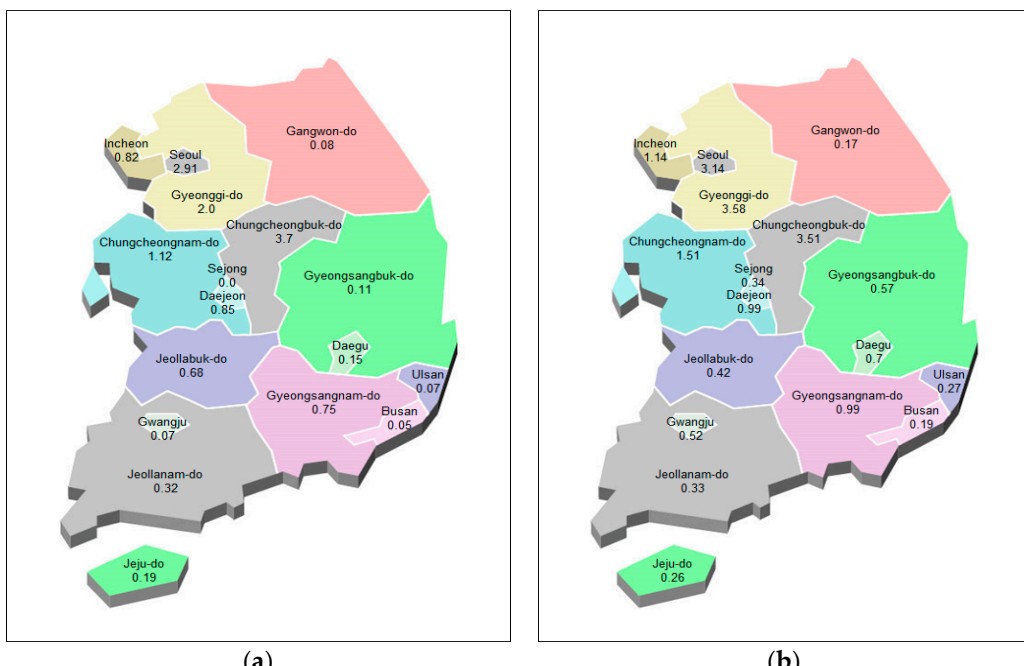

**Figure 12.** National prediction map of epidemic of hepatitis A. (**a**): actual values, (**b**): predicted values.

## 4. Conclusions

In this paper, we propose a prediction model for the epidemic of hepatitis A. We analyzed the correlation between environmental factors and hepatitis A based on data collected from the public data system in Korea. The predictions of the area of occurrence were performed based on 3D LSTM, a machine learning method, using information on the water environment, the weather, the population, air pollution information, and hepatitis A patients.

The prediction of hepatitis A showed high accuracy with an error of about person per 100,000 population. We confirm that the environmental information in this study can predict the prevalence of hepatitis A. In addition, our study confirmed that fecal coliform count and PM10 among the environmental information were factors of high importance in

predicting hepatitis A. In the future research, we will identify factors that increase reliability and apply them to more infectious diseases.

**Author Contributions:** Conceptualization, M.L. and I.N.; data curation, K.L., M.L. and I.N.; formal analysis, K.L. and I.N.; funding acquisition, I.N.; methodology, K.L., M.L. and I.N.; project administration, I.N.; resources, K.L. and M.L.; supervision, I.N.; validation, M.L. and I.N.; visualization, M.L.; writing—original draft, M.L. and I.N.; writing—review and editing, K.L. and I.N. All authors have read and agreed to the published version of the manuscript.

**Funding:** This study was supported by research fund from Chosun University, 2020.

**Data Availability Statement:** We used public data from KCDC to study only information on the number of cases of infectious diseases by region. 1. Patient information: KCDC (Korea Centers for Disease Control & Prevention), http://www.cdc.go.kr; 2. Water Environment Information: National Institute of Environmental Research (NIER), http://water.nier.go.kr/publicMain/mainContent.do; 3. Weather Information: KMA (Korea Meteorological Agency), https://data.kma.go.kr; 4. Population Information: Public Open Data Portal, https://www.data.go.kr/; 5. Air Pollution Information: KECO (Korea Environment Corporation) AirKorea, https://www.airkorea.or.kr.

**Conflicts of Interest:** The authors declare no conflict of interest.

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
