# Peer review of "Predicting Regional Outbreaks of Hepatitis A Using 3D LSTM and Open Data in Korea"

_electronics, doi:10.3390/electronics10212668_

Round 1

Reviewer 1 Report

The use of statistical methods, machine learning, deep learning, and open data to predict outbreaks is certainly critical, as the recent pandemic has shown us. 
The use of available information and standard methods ensures that the approach can also be used in other contexts and for other purposes.
However, there are aspects in the paper that need to be better defined:
* lines 144-154: how was the information in Table 5 obtained? Where are the values in blue and pink in table 5? "We have obtained the 0.7054 in R-squared value. " (line 146) I do not understand how this value was obtained and where it is shown.
* How was the "data scaling" done? why is Total Nitrogen blank in table 7? how are the other values obtained?
* How was the hyperparameter tuning done?  The article states "After tuning the parameters, the best optimal algorithm is Gradient boosting which the value changes from 0.077935 to 0.0759682." (line 201 ) More details should be added.
* How was LSTM implemented, with what parameters?

The article is interesting and useful.
I suggest accepting it with minor revision.

Translated with www.DeepL.com/Translator (free version)

Reviewer 2 Report

Recommendation: Major Revision

This paper has several shortcomings:

  1. There are short sentences in the prediction system of hepatitis A. Please extend them.
  2. Please explain which method is used for data preprocessing.
  3. Please add the preprocessing equations.
  4. The motivation for the methodology is not clear.
  5. Figure 5、6 difficult to read.
  6. Please cite more recent references from Electronics to show the relevance of your study for the journal.

Reviewer 3 Report

It is an interesting paper in predicting Hepatitis trends. But one biggest problem to stop a publishments is the lack of explanation for most tables and figures.

  1. Sec 3.1 "We replace missing 122 values with new values according to the missing values policy as shown in Table 3(lower)." I am very confused with your missing value policy. Please explain it.
  2. Fig 4b show  integrated living areas. What do different colors stand for?
  3. I didn't find dark blue in Table5. Also, I am curious about items listed in Table5. Are they all factors OR authors highlight partial of them?
  4. Explain Table/Figure 7 since readers know nothing......\
  5. Last paragraph of 3.1, "We found no problem with multi-collinearity........"  stop using "no problem" in a research paper. It will ruin your credibilities.......
  6.  

Round 2

Reviewer 2 Report

Recommendations: Accept

Reviewer 3 Report

good to be published